# Antimicrobial Resistance in Isolates from Cattle with Bovine Respiratory Disease in Bavaria, Germany

**DOI:** 10.3390/antibiotics10121538

**Published:** 2021-12-15

**Authors:** Alexander Melchner, Sarah van de Berg, Nelly Scuda, Andrea Feuerstein, Matthias Hanczaruk, Magdalena Schumacher, Reinhard K. Straubinger, Durdica Marosevic, Julia M. Riehm

**Affiliations:** 1Bavarian Health and Food Safety Authority, 85764 Oberschleissheim, Germany; alexander.melchner@t-online.de (A.M.); Sarah.vandeBerg@lgl.bayern.de (S.v.d.B.); Nelly.Scuda@lgl.bayern.de (N.S.); heubeck.a95@gmail.com (A.F.); Matthias.Hanczaruk@lgl.bayern.de (M.H.); Magdalena.Schumacher@lgl.bayern.de (M.S.); Durdica.Marosevic@lgl.bayern.de (D.M.); 2Institute of Infectious Diseases and Zoonoses, Department of Veterinary Sciences, Faculty of Veterinary Medicine, Ludwig-Maximilians-University, 80539 Munich, Germany; Reinhard.Straubinger@micro.vetmed.uni-muenchen.de

**Keywords:** bovine respiratory disease, antimicrobial resistance, multidrug-resistance, *Pasteurella multocida*, *Mannheimia haemolytica*, *Truperella pyogenes*, dairy farm

## Abstract

Patterns of antimicrobial resistance (AMR) regarding *Pasteurella multocida* (n = 345), *Mannheimia haemolytica* (n = 273), *Truperella pyogenes* (n = 119), and *Bibersteinia trehalosi* (n = 17) isolated from calves, cattle and dairy cows with putative bovine respiratory disease syndrome were determined. The aim of this study was to investigate temporal trends in AMR and the influence of epidemiological parameters for the geographic origin in Bavaria, Germany, between July 2015 and June 2020. Spectinomycin was the only antimicrobial agent with a significant decrease regarding not susceptible isolates within the study period (*P. multocida* 88.89% to 67.82%, *M. haemolytica* 90.24% to 68.00%). Regarding *P. multocida*, significant increasing rates of not susceptible isolates were found for the antimicrobials tulathromycin (5.56% to 26.44%) and tetracycline (18.52% to 57.47%). The proportions of multidrug-resistant (MDR) *P. multocida* isolates (n = 48) increased significantly from 3.70% to 22.90%. The proportions of MDR *M. haemolytica* and *P. multocida* isolates (n = 62) were significantly higher in fattening farms (14.92%) compared to dairy farms (3.29%) and also significantly higher on farms with more than 300 animals (19.49%) compared to farms with 100 animals or less (6.92%). The data underline the importance of the epidemiological farm characteristics, here farm type and herd size regarding the investigation of AMR.

## 1. Introduction

Bovine respiratory disease (BRD) is one of the most significant health problems in bovine medicine worldwide [1]. The syndrome causes significant economic losses in both beef and dairy production farms [2,3]. Regarding its impact on US feedlots, BRD is the most important disease, with an annual incidence of up to 44%, resulting in economic losses of 13.90 USD per animal regarding treatment costs and lower weight gains [3]. Preweaned calves are most affected by BRD in dairy farms [2]. Furthermore, the pregnancy rates, milk yield, and longevity of dairy cows are also negatively influenced by this syndrome [4,5,6]. The etiology of BRD is multifactorial, as it is caused by infectious and non-infectious factors [7,8]. Stressful conditions are involved in the development of BRD, such as commingling of calves from different sources or transports over long distances [7,9]. Further, viral agents, such as bovine parainfluenza virus type 3 (PI-3), bovine respiratory syncytial virus (BRSV), bovine herpes virus type 1 (BHV-1), bovine viral diarrhea virus (BVDV) and bovine coronavirus (BCoV) are associated with BRD and may promote secondary bacterial infections by impairing the animals’ immune system [8,10,11,12]. Lastly, bacterial pathogens, such as *Mannheimia haemolytica*, *Pasteurella multocida*, *Bibersteinia trehalosi*, *Histophilus somni*, *Mycoplasma bovis* and *Truperella pyogenes* contribute to the clinical picture [8,10,13,14]. These may cause various forms of pneumonia with an acute, subacute, or chronic course. The different forms of disease representation include mainly fibrinous pleuropneumonia, which is the most common form of acute pneumonia in weaned, stressed beef cattle, and suppurative bronchopneumonia often seen in young dairy calves [15,16].

Suitable preventive measures start from the management of young calves and comprise an adequate colostrum supply [17]. Optimized housing conditions with appropriate ventilation that provide adequate air exchange also show preventive effects on BRD [18,19,20]. Vaccination against both bacterial and viral pathogens is a valuable prevention measure and leads to improved animal health and fewer economic losses [21,22,23]. However, antibiotic treatment is indicated for controlling acute bacterial infection and the emerged BRD syndrome [23]. In Germany, the approved classes of antibiotic agents aiming at the treatment of respiratory diseases with bacterial origin include ß-lactam antibiotics, fluoroquinolones, phenicols, tetracyclines, trimethoprim-sulphonamides, aminoglycosides, lincosamides, and macrolides, respectively [24]. Besides the treatment of individual diseased animals, metaphylactic medication of all animals within one epidemiological flock is important in this context [25,26,27]. Metaphylaxis may include antibiotic treatment of clinically healthy animals, if they had close contact with already infected animals, as these are likely to be infected [28].

Worldwide studies indicate that there is a trend towards increasing bacterial resistance towards certain antimicrobial agents, especially multidrug-resistance (MDR) when pathogens of BRD are investigated [29,30,31,32,33,34,35,36,37]. In the context of the BRD complex, *P. multocida* and *M. haemolytica* isolates are categorized as MDR if they are not susceptible (resistant or intermediate) to at least one agent in at least three antimicrobial classes [38]. Exposure, overuse, or even misuse of antimicrobial substances do provide evolutionary advantages and may result in resistant bacteria [39,40,41]. Resistance genes spread between pathogens from the bovine respiratory tract. Two forms of horizontal gene transfer appear to play a key role, namely plasmids and so-called integrative and conjugative elements (ICEs) [30,33,42]. The latter contains an entire collection of resistance genes that may transfer horizontally within one single event between strains, species, and even different bacterial genera [43,44,45].

The alarming increase of antimicrobial resistance (AMR) in both, human and veterinary medicine, as well as the fact that antimicrobial resistant strains do circulate between humans and animals, set the impulse for the World Health Organization (WHO) to adopt a global action plan against increasing antimicrobial resistance in 2015 [46,47]. The primary goal was to ensure that the treatment and therapy of infectious diseases in both human and veterinary medicine will remain effective in the future. Therefore, WHO statements plead for responsible and prudent use of antimicrobial substances [46]. In Germany, this global action plan of the WHO was implemented in the so-called German Antibiotic Resistance Strategy (DARTS) in 2008, and thoroughly followed since then. Important elements of this strategy were the establishment of monitoring systems for the detection of AMR as well as new legal regulations, such as the documentation of antibiotic consumption levels, the determination of therapy frequency in fattening farms, as well as the obligation of antimicrobial resistance testing under certain conditions for veterinarians [48,49].

The aim of this study was to complement the already existing resistance monitoring programs, to record current trends in the development of AMR and MDR with regard to bacterial pathogens of BRD in Bavaria over the last 5 years and finally to derive treatment recommendations from this. Furthermore, the influence of epidemiological parameters, such as farm type and farm size on the resistance pattern were investigated.

## 2. Results

### 2.1. Bacterial Isolates

Between July 2015 and June 2020, a total of 754 isolates were collected from 662 animals with suspected BRD syndrome, origination from 519 farms were included in the present study. *P. multocida* was the most frequently isolated pathogen with 345 (45.76%), followed by 273 *M. haemolytica* (36.21%), 119 *T. pyogenes* (15.78%), and 17 *B. trehalosi* (2.25%) isolates (Table 1 and Figure 1).

### 2.2. Five-Year Antimicrobial Susceptibility

Low resistance rates with a proportion of not susceptible isolates of less than five percent were found for *P. multocida* isolates (n = 345) in the case of cephalosporin class (ceftiofur), penicillin class (penicillin G), phenicol class (florfenicol), and fluoroquinolone class (enrofloxacin) (Table 2 and Appendix A). The fraction of not susceptible isolates was higher for the macrolide antibiotic tulathromycin (15.65%) (Table 2 and Appendix A). The highest proportion of not susceptible *P. multocida* isolates was found for tetracycline (39.42%), and spectinomycin (78.84%) (Table 2 and Appendix A).

The proportion of not-susceptible *M. haemolytica* isolates (n = 273) collected over the five-year range was below five percent for ceftiofur, for penicillin G, for enrofloxacin, for florfenicol, and for tulathromycin. It was slightly higher for the macrolide compound tilmicosin with 6.59% (Table 2 and Appendix A). The highest not susceptibility rates were found when isolates were tested with tetracycline (21.25%), and the aminocyclitol class compound spectinomycin, 80.95% (Table 2 and Appendix A).

For *T. pyogenes* isolates (n = 119) and *B. trehalosi* isolates (n = 17) no defined species-specific minimum inhibitory concentration (MIC) breakpoints according to CLSI VET guidelines are available to categorize these into susceptible and not susceptible (intermediate and resistant) [50,51,52]. The distribution of MIC values of these two pathogens is shown in Appendix A.

### 2.3. Trends in Not Susceptibility

The trend analysis of the annual not susceptibility rates pertaining to the species *P. multocida* and *M. haemolytica* revealed a decreasing tendency only for the aminocyclitol agent spectinomycin (Table 3 and Appendix A, Figure 2a,b). The proportion of not susceptible isolates regarding *P. multocida* isolates decreased from 88.89% in the first study year (July 2015 to June 2016) to 67.82% in the last study year (July 2019 to June 2020; OR = 0.70; 95% CI: 0.56–0.86; *p* < 0.001). Regarding *M. haemolytica* isolates it decreased from 90.24% in the first study year to 68.00% in the last study year (OR = 0.71; 95% CI: 0.55–0.90; *p* = 0.005; Table 3 and Appendix A; Figure 2a,b). For the investigated *P. multocida* isolates significantly increasing rates of not susceptible isolates were found within the study period for the antimicrobial agents tulathromycin (5.56% to 26.44%; OR = 1.60; 95% CI: 1.25–2.08; *p* < 0.001) and tetracycline (18.52% to 57.47%; OR = 1.62; 95% CI: 1.36–1.94; *p* < 0.001; Table 3 and Appendix A; Figure 2a).

### 2.4. Multidrug-Resistance

In veterinary medicine, *P. multocida* and *M. haemolytica* isolates of BRD are classified as multidrug-resistant (MDR) if they are not susceptible to at least one agent in at least three antimicrobial classes [38]. Following this definition, the prevalence of MDR *P. multocida* and *M. haemolytica* isolates was determined in this study. The eight antibiotic agents penicillin G, ceftiofur, florfenicol, enrofloxacin, tilmicosin (only for *M. haemolytica*), tulathromycin, tetracycline and spectinomycin from the seven antimicrobial classes penicillins, cephalosporins, phenicols, fluoroquinolones, macrolides, tetracyclines, and aminocyclitols were included in this MDR analysis. The highest proportion of MDR-isolates was found for *P. multocida* (13.91%), whereas of the *M. haemolytica* isolates only 5.13% were categorized as MDR (Table 4, Appendix A). The analysis of annual MDR rates of the bacterial pathogens showed a significant increase over the five-year period for *P. multocida* from 3.70% (first year) to 22.99% (final year) (OR = 1.61; 95% CI: 1.25–2.14; *p* < 0.001) (Figure 3 and Appendix A).

### 2.5. Additional Epidemiological Investigations

Further epidemiological investigations were carried out including the 618 MDR *P. multocida* and *M. haemolytica* isolates. Information on the distribution of animal and farm characteristics is displayed in Appendix A. Most isolates originated from male animals (56.63%), one to two months old (34.95%) and diseased due to BRD (44.98%). PI-3, *Mycoplasma* species and BRSV were detected in 4.05%, 15.37% and 12.94% of the isolates. Most isolates are derived from farms in Upper Bavaria (30.58%), with 101 to 300 animals (58.41%), fattening farms (50.97%) and farms with a therapy frequency of five or less (20.06%).

The results of the univariable and multivariable logistic regression to determine the association of occurrence of MDR *P. multocida* and *M. haemolytica* isolates with certain farm or animal characteristics are shown in Appendix A. Regarding the individual animal characteristics, neither sex and age nor the detection of *M. bovis* or PI-3 and BRSV were statistically significantly associated with the occurrence of MDR (Appendix A). Additionally, the odds of the occurrence of MDR were not significantly higher among animals, which had died due to the BRD complex as compared to animals that had survived the disease (Appendix A). Among the farm characteristics, neither the geographical location in one of the seven administrative districts nor the farm antibiotic therapy frequency was statistically significantly associated with the occurrence of MDR isolates (Appendix A). There was a significant association between the occurrence of MDR *P. multocida* and *M. haemolytica* isolates and the size of a farm (Figure 4a). In farms with more than 300 animals, the odds for MDR isolates were significantly higher as compared to farms with a size of 100 animals or less (Adjusted OR = 2.89; 95% CI: 1.26–7.29; *p* = 0.017; Appendix A). Our analysis showed that in farms with 100 animals or less, 6.92% of all isolates were MDR, on farms with 101 to 300 animals 8.31% were MDR, while on farms with more than 300 animals 19.49% of all isolates were MDR (Figure 4a).

In addition, the odds for isolating MDR isolates were significantly lower in dairy farms (aOR = 0.23; 95% CI: 0.08–0.54; *p* = 0.002) and mixed farms (aOR = 0.46; 95% CI: 0.20–0.93; *p* = 0.042) as compared to in pure fattening farms (Appendix A). Only 3.29% of isolates in dairy farms were MDR, 6.52% were MDR in mixed farms, while in fattening farms 14.92% were MDR (Figure 4b).

## 3. Discussion

### 3.1. New Legal Regulations and Increase in Tested Isolates

As a response to the increasing trend in the emergence of resistant pathogens and WHO’s global action plan on AMR, the German Antibiotic Resistance Strategy (DARTS) was developed and numerous legal changes have been made [46,48]. Out of these, the amendment to the 2018 “Tierärztliche Hausapothekenverordnung”, a national German law, obliges veterinarians to ensure the efficacy regarding antimicrobial therapy applying prior resistance testing under certain conditions [49]. This legal change is visible, in our study, as the number of all tested bacterial isolates sent to our laboratory has increased since the third investigation year with 151 isolates compared to 197 isolates in the following observation period (Table 1).

### 3.2. Therapy Guide for the Practitioner

With AMR of bacterial isolates on the rise, precise knowledge of the resistance situation at hand is essential for the targeted treatment of bacterial infections [29,30,32,34]. Since there is only an obligation to determine resistance in certain cases and antibiotic therapy must be started immediately in acute cases of the disease, the practicing veterinarian has to rely on existing data and studies on the local resistance situation [49,53]. Currently, the Germany-wide resistance monitoring program GERM-Vet as well as the Swiss therapy guide for veterinarians advises valuable treatment recommendations considering pharmacological aspects [34,54]. The present study was evaluated on reappraising these guidelines with up-to-date clinical data from Bavaria, Germany (Table 2).

### 3.3. Antimicrobial Agents with a Favourable Resistance Situation

The Swiss therapy guideline recommends the phenicol agent florfenicol as a first-line antibiotic for the treatment of acutely ill animals with BRD [54]. This compound offers several advantages: firstly, it has a bactericidal effect and thus has advantages over bacteriostatic agents that only inhibit growth and replication and thus depend on good immunocompetence, which may no longer be present in cattle suffering from BRD [53,54,55,56]. Secondly, one shot preparations are approved and very practical to use, as a single subcutaneous injection is sufficient [24]. In our analysis, florfenicol showed excellent efficacies for all pathogens (Table 2). However, nine of 14 florfenicol not susceptible *P. multocida* isolates were isolated in the final study year, which might indicate a tendency of upcoming resistance of this bacterial species against florfenicol and needs to be observed in detail in the future (Appendix A). Similar increasing trends towards florfenicol-resistant *M. haemolytica* have been reported by the GERM-Vet data in recent years. That underlines the importance of continuous monitoring of resistance trends [34]. We conclude that the benefits of florfenicol outweigh the above-mentioned upcoming risk of resistance development and still recommend florfenicol for therapy in cattle suffering from BRD.

As second-line antibiotics, the Swiss therapy guideline recommends, among others, the use of all compounds of the ß-lactam antimicrobials, with the exception of third-generation cephalosporins, as ceftiofur [54]. In the present study, the proportion of not susceptible isolates for penicillin G regarding the investigated bacterial species was below five percent (Table 2). Consequently, we recommend penicillin G, a member of the ß-lactams, for the therapy of BRD.

At the third-line position, the Swiss therapy guide lists the 3rd and 4th generation cephalosporins, which include the antimicrobial agent ceftiofur and the fluoroquinolone antimicrobial class, including enrofloxacin [54]. It should be noted that these represent important therapeutic reserve antibiotics in human medicine against (MDR) germs, such as methicillin/oxacillin-resistant staphylococci. The WHO classifies these substances, therefore, as “highest priority critically important antimicrobials” and calls on veterinarians to use them prudently, if at all necessary [53,57]. In Germany, too, the cephalosporins of the 3rd and 4th generation as well as the fluoroquinolones are classified as reserve antibiotics and should only be used as antibiotics of last resort if no other effective compounds are available [53]. The legislator, therefore, requires resistance testing or a known resistance situation in the respective farm, which is known from previous antimicrobial susceptibility testing, for every use of these reserve antibiotics [49,53]. Given that ceftiofur is an antimicrobial of last resort, it is quite encouraging that the fraction of not susceptible isolates for *P. multocida* and *M. haemolytica* in our study was less than one percent (Table 2). Data originating from a current North American study regarding feedlot cattle show that the resistance rates for ceftiofur, which are between 3% and 33% for *P. multocida* and between 0% and 4% for *M. haemolytica*, were substantially higher compared to our study from Bavaria (Table 2) [30]. The outcome of the resistance situation regarding the fluoroquinolone agent enrofloxacin is also favorable (Table 2). Given that both ceftiofur and enrofloxacin are important therapeutic reserves, these two antimicrobial agents can only be recommended for therapy to a limited and well-considered extent (Table 2).

### 3.4. Antimicrobial Agents with Unfavorable Resistance Situation

In the Swiss therapy guidelines, the agent tetracycline is mentioned as a second-line antibiotic for the treatment of acutely ill animals and as an antimicrobial substance for metaphylaxis. However, it is pointed out that its efficacy is limited due to a considerable AMR rate [54]. The latter has been described for BRD pathogens already since the 1990s and was confirmed since then in studies from all over the world [29,30,34,35]. The unfavorable resistance situation was also reflected in the present study. 39.42% of *P. multocida* and 21.25% of *M. haemolytica* isolates were revealed to be not susceptible (Table 2). For *P. multocida*, a significant increase of the portion of not susceptible isolates from 18.52% in the first study year up to 57.47% in the last study year was found (Table 3 and Appendix A, Figure 2a). Reasons for the decrease in efficacy could be the high use of this compound. In North American feedlots, tetracycline is one of the most frequently used antibiotics for the treatment of BRD, but also for the prevention of liver abscesses [30,31]. In Germany, tetracycline is in terms of volume the most frequently used antibiotic for calves and cattle kept in fattening farms [58]. Due to the demonstrated increase in resistance levels and very poor efficacy, the general use of tetracycline in calf and cattle fattening should be reconsidered and can, therefore, not be recommended for the therapy of BRD (Table 2).

Antibiotics from the macrolide class are also listed as agents for metaphylactic treatment in the Swiss therapy guidelines [54]. We observed a significant increase in not susceptible *P. multocida* isolates from 5.56% in the first study year up to 26.44% in the last study year regarding tulathromycin (Table 3 and Appendix A, Figure 2a). Within the scope of the Germany-wide resistance monitoring GERM-Vet, an increase of resistant *P. multocida* isolates from 3% in 2016 up to 14% in 2018 was also detected and currently confirms the trend towards a higher resistance rate against tulathromycin [34]. It is of particular concern that although this compound has been authorized in Europe by the European Medicines Agency only since 2003, its resistance situation has increased so rapidly within only few years [59]. This fact is furthermore worrying, as tulathromycin is not only approved and used for therapy but also for metaphylactic treatment [24,30,33]. There is a strong accumulation in inflamed lung tissue and also accomplishes a concentration above the minimum inhibitory concentration (MIC) of over seven days after a single subcutaneous injection [24,60]. Such one-shot preparations, therefore, offer enormous advantages purely from a hands-on point of view, since an animal only needs to be treated once at a time. Nevertheless, it should be emphasized that this antimicrobial class also belongs to the “highest priority critically important antimicrobials” defined by the WHO and represent one of a few available therapeutic options for serious bacterial infections [57]. In this context, the use of tulathromycin in metaphylaxis should also be reconsidered (Table 2).

Spectinomycin from the aminocyclitol class cannot be recommended for the treatment of BRD due to an unfavorable resistance situation (Table 2). With a proportion of 78.84% not susceptible *P. multocida* and 80.95 % not susceptible *M. haemolytica* isolates, spectinomycin represents the antimicrobial agent with the highest proportion of not susceptible isolates in our analysis (Table 2). Moreover, spectinomycin is not among the most frequently used compounds in calf and cattle fattening, and the consumption quantities did not increase in recent years [30,58]. However, a significant decrease of not susceptible isolates against spectinomycin from 88.89% to 67.82% and 90.24% to 68.00% could be seen in *P. multocida* and *M. haemolytica* isolates (Table 3 and Appendix A, Figure 2a,b).

### 3.5. Multidrug-Resistance

As described above, eight antibiotic agents from seven antimicrobial classes were included in the MDR analysis because they have species-specific breakpoints according to the CLSI VET guidelines [50]. However, it must be mentioned that there are other antibiotic agents from these seven classes with species-specific minimum inhibitory concentration (MIC) breakpoints for respiratory diseases in cattle. For example, the agent tildipirosin and gamithromycin from the macrolide class or danofloxacin from the fluoroquinolone class were not tested for susceptibility in our laboratory despite the presence of species-specific breakpoints and thus could not be included in the MDR analysis. Ampicillin could also not be included in the analysis because the inhibitory concentrations on the microtiter plate we used were in a higher range than the breakpoints set by CLSI [50]. Since we did not test all antibiotic agents with defined breakpoints for susceptibility, it must be assumed that even more isolates in our data set could be characterized as MDR. A notable finding in our study was the higher rate of MDR *P. multocida* isolates (13.91%) compared to *M. haemolytica* isolates (5.13%) (Table 4; Appendix A). This effect was explained in prior publications by different gene transfer and integration rates, or the persistence of ICEs hosting resistance genes and regarding diverse bacterial species [30]. However, the rate of MDR *P. multocida* isolates increased significantly from 3.70% in 2015/2016 to 22.99% in 2019/2020 in the present study (Figure 3, Appendix A). This most alarming result was observed also in isolates from North America and illustrates that increasing AMR is a worldwide problem [30,37]. Resistance levels in North America appear high, with proportions of MDR *P. multocida* isolates exceeding 90% and proportions of MDR *M. haemolytica* isolates exceeding 80%, respectively [30]. These numbers exceed those determined in the present study for Bavarian farms (Figure 2, Appendix A). It must be mentioned, however, that it is difficult to compare MDR prevalences from different studies, as MIC breakpoints other than those specific to veterinary medicine are often used to divide isolates into susceptible, intermediate and resistant [38].

### 3.6. Additional Epidemiological Investigations

The investigation of further epidemiological parameters concluded that no animal characteristics were associated with a higher probability of occurrence of MDR *P. multocida* and *M. haemolytica* isolates (Appendix A). However, it was seen that the odds for MDR isolates were significantly lower in dairy farms (aOR = 0.23; 95% CI: 0.08–0.54; *p* = 0.002) and mixed farms (aOR = 0.46; 95% CI: 0.20–0.93; *p* = 0.042) compared to fattening farms (Appendix A, Figure 4b). The reasons why the resistance problem mainly affects fattening farms can only be speculated and requires further research. However, it is known that the stressful transport from a dairy farm, birthplace, to the fattening farm, as well as the assortment of calves from many individual farms of origin, increases the risk of BRD and thus the need for antimicrobial treatment [7,9,23]. In the U.S., such groups of animals at increased risk for BRD are treated metaphylactically upon arrival in the feedlot to reduce morbidity and mortality rates and achieve better fattening results [25,26,27,30,31]. In Germany, metaphylactic treatment of an entire group of animals is also permitted. However, it requires diseased animals within this group that show clinical signs and the concern that the healthy animals in the group will also rapidly become ill [28,53]. In prior studies investigating the metaphylactic use of antimicrobial agents in groups of animals, it was shown that the administration of antimicrobial agents favored the shedding of MDR isolates and increased the likelihood of finding such MDR isolates in stablemates after contagious spreading [61,62]. Equally important to mention in this context is the small farm structure of dairy farms within Bavaria with an average herd size of 40 dairy cows per farm [63]. On these farms, calves are often kept individually in calf hutches during the first weeks of life. On the one hand, this is associated with a lower risk of developing BRD and possibly results in a more targeted individual antimicrobial treatment for various diseases compared to the situation in fattening farms [64,65,66]. There, the beef cattle are kept in groups and subsequently treated possibly as an epidemiologic unit [25,26,27,30,31]. In order to limit antimicrobial metaphylaxis and the resulting development of AMR, German law requires laboratory diagnostics including pathogen identification and AMR testing in the case of repeated use of antibiotics in certain age groups and production steps [49].

In addition to the type of farm, the size of the farm is also a critical variable (Figure 4a). In the present study, the odds for MDR isolates were significantly higher on farms with more than 300 animals than on farms with 100 animals or less (aOR = 2.89; 95% CI: 1.26–7.29; *p* = 0.017; Appendix A, Figure 4a). At the same time, data from the Federal Ministry of Agriculture and Food show that the frequency of antimicrobial treatment in recent years has been higher for farms with a larger number of animals, and thus more frequently treated with antimicrobials, than on farms with a smaller number of animals [58]. It remains speculative why antimicrobials are used more frequently on farms with a higher number of animals and whether this influenced the higher probability of the presence of MDR isolates. One possible explanation could be that farms with smaller animal numbers have better control of infectious diseases resulting in better individual animal treatment [66]. Other studies have shown that a smaller number of individual animals per group in the animal husbandry departments is advantageous, as the risk of BRD infection increases with the number of animals per group [64,65]. 

In the present study, there was no statistically significant association between the frequency of therapy and the occurrence of MDR isolates (Appendix A). It needs to be mentioned, however, that the values of the treatment frequency only refer to the respective half-year of sampling, but the fattening period lasts more than six months and the values in the preceding or following half-year could differ markedly from the one considered in the analysis. Furthermore, the treatment frequency refers to the entire farm, so it is possible that the animals in our analysis were kept in a barn compartment where fewer antimicrobials were applied. In addition, the treatment frequency refers to all antimicrobials used in the half-year and thus also includes treatments against other diseases. The value of the farm treatment frequency in our study is, therefore, maybe less suitable as an indicator of antimicrobial consumption.

### 3.7. Limits of the Study

The samples analyzed in the study include samples from the upper respiratory tract, such as nasal swabs but also samples from the lower respiratory tract, such as organ samples from necropsy or bronchoalveolar lavage fluid. However, there is evidence that cultures of nasal swabs from the upper airways are not representative of the pathogen in the lower airways [67]. One study revealed that although samples from both the upper and lower airways were positive for *M. haemolytica*, only 77% showed an identical pulse field gel electrophoresis type [67]. We cannot rule out that the isolates originating from nasal swabs in our study may not be responsible for the clinical picture of BRD.

Another disadvantage of the study is that it is not known whether or how often antimicrobial treatment was applied before sampling. The extent to which immediate antimicrobial treatment before sampling influences the resistance pattern is also controversially discussed in other studies [30,33,61,62]. However, it could be that a previous antimicrobial treatment exerts a selection pressure towards more resistant strains and that the original microbial flora is not represented in these samples.

Another important point to mention is that our study is not an analysis with a clearly defined sampling plan, as is the case, for example, in the national resistance monitoring GERM-Vet, but is a retrospective evaluation of all isolates sent in [34]. Therefore, and following previous publications, only a single individual of each species was included per quarter of a year per farm in our analysis to prevent bias and overrepresentation of clonal isolates [36,68]. Nevertheless, there could also be a potential geographical bias in our dataset, as described in other studies, because our study only includes isolates from Bavaria, a single state of Germany, and even within Bavaria, more samples in our analysis originate from the southern districts than from the northern ones (Appendix A) [30,36,37].

Finally, it should be mentioned that additional molecular screening for AMR genes, as also carried out in recent publications, could provide further insights, especially with regard to the role of ICEs in the spread of MDR isolates and should be part of future endeavors [30,33].

## 4. Materials and Methods

### 4.1. Study Design and Origin of Animals

Data included in the present study were collected within the scope of the state veterinary laboratory diagnostics at the Bavarian Health and Food Safety Authority. In the present study, the investigated samples originated from calves, cattle, or dairy cows with putative symptoms of BRD in Bavaria, Germany, from July 2015 to June 2020. In the present study, cows were kept in dairy farms solely for the purpose of milk production, in fattening farms, cattle were kept for meat production, and finally, in mixed farms, both categories of animals were kept. In order to prevent bias and over-representation of clonal isolates, only one isolate of a species per farm per quarter year was included in the data set, following previous publications [36,68].

### 4.2. Bacterial Isolates

The specimens, here nasal swabs, bronchoalveolar lavage fluid, or lung tissue samples, were analyzed in the ISO 17025 accredited laboratory at the Bavarian Health and Food Safety Authority. Samples were initially inoculated on Columbia sheep blood agar (Oxoid, Wesel, Germany) and incubated at 37 °C for 24 to 48 h under aerobic conditions as well as under a microaerophilic atmosphere, at 10% CO_2_. To isolate pure suspicious colonies of *P. multocida*, *M. haemolytica*, *B. trehalosi* or *T. pyogenes*, fresh subcultures were incubated under the above-described conditions. Identification of bacterial species was carried out using MALDI-TOF MS (Bruker, Bremen, Germany).

Regarding the isolation of *Mycoplasma* species, animal samples were inoculated in specific Thermo Scientific™ Mycoplasma/Ureaplasma Broth that inhibits the growth of most gram-negative, gram-positive bacteria, as well as yeasts (Thermo Scientific, Schwerte, Germany), and incubated microaerophilic for 120 h at 37 °C with 10% CO_2_.

### 4.3. Antimicrobial Susceptibility Testing

Antimicrobial susceptibility testing was carried out according to the protocols published in VET01 5th edition, VET01S 5th edition and VET06 1st edition, by the Clinical and Laboratory Standards Institute (CLSI), Wayne, PA, USA [50,51,52]. The microbroth dilution method was carried out on 16 different antibiotic substances as commercially available and according to the manufacturer’s instructions (Micronaut-S, Grosstiere 4, Merlin, Bruker, Bornheim, Germany). This panel was designed to test on recommended antibiotics for the treatment of farm animals in Germany. The minimum inhibitory concentration (MIC) of each isolate and antimicrobial substance was metered using a photometric plate reader system (Micronaut scan, MCN6 software, Merlin, Bruker, Bornheim, Germany). Subsequently, the MIC value was reconciled with determined species-specific breakpoints to categorize the respective *M. haemolytica* and *P. multocida* isolates into “susceptible”, “intermediate” and “resistant” for the tested antimicrobial agents: ceftiofur, penicillin G, florfenicol, enrofloxacin, tilmicosin (only *M. haemolytica*), tulathromycin, tetracyclin and spectinomycin [50]. For the antibiotic agent amoxicillin clavulanic acid, cephalotin, trimethoprim-sulfamethoxazole, colistin, tiamulin, erythromycin and gentamicin, no species-specific breakpoints for bovines with BRD are available. Specific breakpoints for *T. pyogenes* and *B. trehalosi* have not been published by the CLSI for any of the tested antibiotic agents for veterinary medicine either, so that only the distribution of the MIC can be given [50,51,52].

Regarding the BRD syndrome, *P. multocida* and *M. haemolytica* were termed MDR isolates if they were not susceptible (intermediate and resistant) to at least one antibiotic substance in three or more antimicrobial classes [38]. Following this definition, our study investigated the prevalence of MDR *P. multocida* and *M. haemolytica* isolates under epidemiological aspects.

### 4.4. Viral Isolates

Within the scope of the diagnostic services at the Bavarian Health and Food Safety Authority, Germany, results on further viral pathogens were incorporated regarding BRSV and PI-3.

### 4.5. Epidemiological Data

In addition to the isolated pathogens and respective resistance, epidemiological data on the isolated was collected, including sex of the animal, age of the animal, geographical location of the farm, type of farm, herd size of the farm and antimicrobial therapy frequency of the farm, respectively. Furthermore, it was investigated whether the animal died because of BRD. Data were obtained from the German database “Herkunftssicherungs- und Informationssystem für Tiere” (HIT). The HIT database contains comprehensive data on every single animal, including date of birth, sex, date of death and the status of animal diseases, such as BHV-1. For reasons of animal traceability, the database minutely reveals dates and addresses of trading procedures. Extra data pertaining to farms, such as the geographical location, the age and sex statistics on herds, the number of animals and the corresponding antimicrobial therapy frequency were also be downloaded. All results on animals were connected to the unique ear tag number that is assigned to each animal in the HIT database. It further allowed linking the respective farm characteristics from the HIT database, even beyond the death of an animal. Death due to BRD was defined as death within 14 days after diagnosis, assuming a median recovery time from BRD of 14 days [8]. All data on farms included in the study were determined retrospectively for the initial sampling date. The geographical location of the farm was extracted on administrative district level in Bavaria, here, North Bavaria (Upper, Middle, Lower Franconia and Upper Palatinate), Lower Bavaria, Upper Bavaria, or Swabia. The classification into the type of farm was made by us on the basis of the age and gender statistics in the HIT database. A farm was defined as a dairy farm if it had female animals with calving and male animals only up to the age of four months. If male animals over four months of age were recorded in addition to female animals with calving, we assumed that this farm with cows and female offspring also kept male animals for fattening and, therefore, the farm is categorized as a mixed farm with milk production and beef production. Farms were defined as fattening farms if they kept only male animals or female animals that had not reached first calving age and were, therefore, not used for milk production. Therapy frequency per half-year represents an indicator of the use of antibiotics. It is calculated by multiplying the number of animals treated by the number of treatment days for each active substance used. The sum of all these multiplications per half-year is then divided by the average number of animals kept in the corresponding half-year. In Germany, this parameter is notified officially regarding fattening farms with more than 20 animals since the 16th Amendment to the Medicinal Products Act in 2014 [58,69].

### 4.6. Statistical Analysis

First, the proportion of isolates containing *M. haemolytica* and *P. multocida*, respectively, per year and for the whole study period was determined. Next, the proportion of not susceptible/MDR isolates was calculated. To investigate whether to proportion of not susceptible/MDR isolates changed over the course of the study period, univariable logistic regression analyses were conducted using the year of sampling as an independent variable. To determine what animal and farm factors are associated with MDR, we conducted multivariable logistic regression analyses. Therefore, the univariable effects of the year the sample was taken, the presence of other pathogens in the isolate (*Mycoplasma* species, BRSV and PI-3), age, sex and disease outcome (diseased vs. deceased) of the animal, as well as region, type (dairy vs. fattening farm), size and therapy frequency of the farm were assessed. Factors with a *p*-value ≥ 0.2 were considered for the multivariable model. The most parsimonious model was determined in a stepwise, forward-selection process. All analyses were conducted in R Statistical Software (R Core Team, R: A language and environment for statistical computing. R Foundation for Statistical Computing, Vienna, Austria, 2021).

## Figures and Tables

**Figure 1 antibiotics-10-01538-f001:**
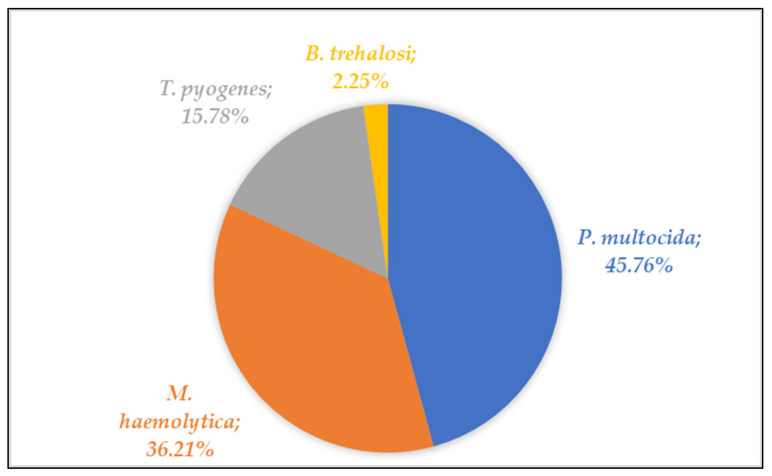
Overall proportion (%) of pathogens detected among the total number of analyzed samples.

**Figure 2 antibiotics-10-01538-f002:**
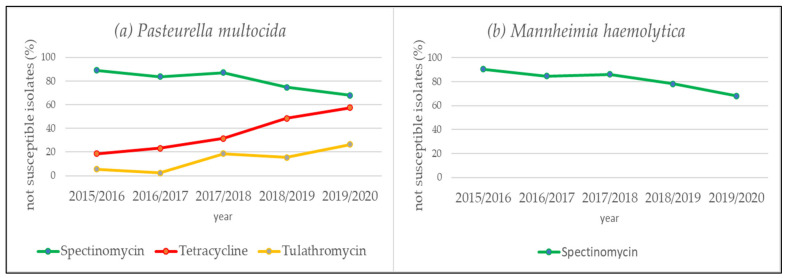
Statistically significant trends regarding the not susceptibility of *P. multocida* (**a**) and *M. haemolytica* (**b**) over the five-year period in Bavaria, Germany. For spectinomycin a significant decrease in not susceptibility could be observed in *P. multocida* (OR = 0.70; 95% CI: 0.56–0.86; *p* < 0.001) (**a**) and in *M. haemolytica* isolates (OR = 0.71; 95% CI: 0.55–0.90; *p* = 0.005) (**b**). For tetracycline (OR = 1.62; 95% CI: 1.36–1.94; *p* < 0.001) and tulathromycin (OR = 1.60; 95% CI: 1.25–2.08; *p* < 0.001) a significant increase in not susceptible *P. multocida* isolates could be observed (**a**).

**Figure 3 antibiotics-10-01538-f003:**
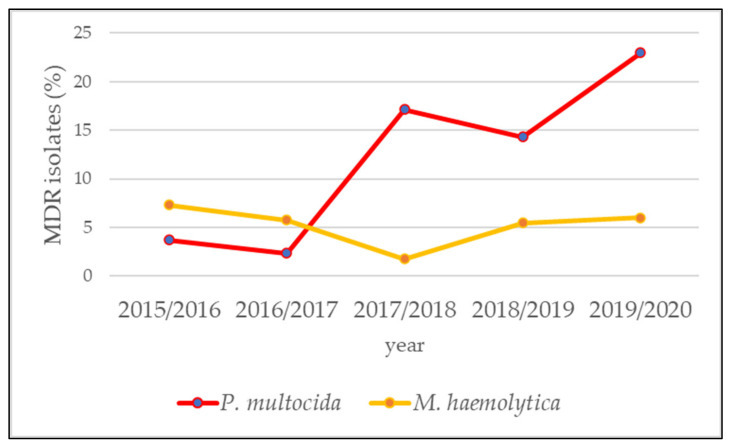
Annual multidrug-resistance (MDR) rates of the bacterial pathogens *P.*
*multocida* and *M. haemolytica* from cattle with bovine respiratory disease (BRD) in Bavaria, Germany. A significant increase of MDR *P. multocida* isolates could be observed over the five-year period 2015–2020 (OR = 1.61; 95% CI: 1.25–2.14; *p* < 0.001).

**Figure 4 antibiotics-10-01538-f004:**
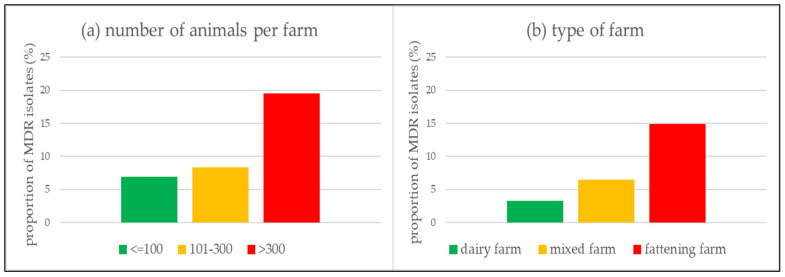
Proportion of multidrug-resistant (MDR) *P. multocida* and *M. haemolytica* isolates depending on farm size (number of animals per farm) (**a**) and type of farm (**b**). In farms with more than 300 animals, the odds for isolating MDR isolates were significantly higher than in farms with 100 or less animals (Adjusted OR = 2.89; 95% CI: 1.26–7.29; *p* = 0.017). In addition, the odds for isolating MDR isolates were significantly lower in dairy (aOR = 0.23; 95% CI: 0.08–0.54; *p* = 0.002) and mixed farms (aOR = 0.46; 95% CI: 0.20–0.93; *p* = 0.042) than in fattening farms.

**Table 1 antibiotics-10-01538-t001:** Species, absolute and (relative) number of isolates investigated in the present study over the five-year period 2015–2020 in Bavaria, Germany.

	2015/2016 n (%)	2016/2017 n (%)	2017/2018 n (%)	2018/2019 n (%)	2019/2020 n (%)	Total
*P. multocida*	54 (49.09)	43 (34.96)	70 (46.36)	91 (46.19)	87 (50.29)	345 (45.76)
*M. haemolytica*	41 (37.27)	52 (42.28)	57 (37.75)	73 (37.06)	50 (28.90)	273 (36.21)
*T. pyogenes*	14 (12.73)	28 (22.76)	21 (13.90)	27 (13.70)	29 (16.76)	119 (15.78)
*B. trehalosi*	1 (0.91)	0 (0.00)	3 (1.99)	6 (3.05)	7 (4.05)	17 (2.25)
Total isolates	110 (100)	123 (100)	151 (100)	197 (100)	173 (100)	754 (100)

**Table 2 antibiotics-10-01538-t002:** Five-year not susceptible rates of bacterial pathogens with defined species-specific breakpoints according to CLSI VET guidelines.

Antimicrobial Class	Antimicrobial Agent	*P. multocida* % (n)	*M. haemolytica* % (n)	Recommendation for Therapy ^1^
cephalosporin	ceftiofur	0.87 (3/345)	0.00 (0/273)	(+/−)
penicillin	penicillin_G	3.48 (12/345)	4.76 (13/273)	(+)
phenicol	florfenicol	4.06 (14/345)	1.10 (3/273)	(+)
fluorochinolone	enrofloxacin	0.29 (1/345)	2.93 (8/273)	(+/−)
macrolide	tilmicosin	no breakpoint ^2^	6.59 (18/273)	(+/−)
	tulathromycin	15.65 (54/345)	2.93 (8/273)	(+/−)
tetracycline	tetracycline	39.42 (136/345)	21.25 (58/273)	(−)
aminocyclitol	spectinomycin	78.84 (272/345)	80.95 (221/273)	(−)

^1^ recommendation for therapy: (+): suitable for therapy, (+/−): partly suitable for therapy, (−): not suitable for therapy; ^2^ no breakpoint according to CLSI VET guidelines.

**Table 3 antibiotics-10-01538-t003:** Statistically significant trends, decrease or increase, regarding the not susceptibility of bacterial pathogens investigated in this study over the five-year period 2015–2020 in Bavaria, Germany.

Pathogen	Antimicrobial Class	Antimicrobial Agent	2015/2016 % (n)	2016/2017 % (n)	2017/2018 % (n)	2018/2019 % (n)	2019/2020 % (n)	OR (95% CI)	*p*-Value
*P. multocida*	Aminocyclitol	Spectinomycin	88.89 (48/54)	83.72 (36/43)	87.14 (61/70)	74.73 (68/91)	67.82 (59/87)	0.70 (0.56–0.86)	<0.001
	Tetracycline	Tetracycline	18.52 (10/54)	23.26 (10/43)	31.43 (22/70)	48.35 (44/91)	57.47 (50/87)	1.62 (1.36–1.94)	<0.001
	Macrolide	Tulathromycin	5.56 (3/54)	2.33 (1/43)	18.57 (13/70)	15.38 (14/91)	26.44 (23/87)	1.60 (1.25–2.08)	<0.001
*M. haemolytica*	Aminocyclitol	Spectinomycin	90.24 (37/41)	84.62 (44/52)	85.96 (49/57)	78.08 (57/73)	68.00 (34/50)	0.71 (0.55–0.90)	=0.005

**Table 4 antibiotics-10-01538-t004:** Amongst the investigated bacterial species, the absolute and (relative) number of isolates was ranked into the characteristic pan-susceptible, if these were susceptible towards all agents tested. Not susceptible isolates revealed to be resistant against at least two tested antimicrobial classes (shaded in light grey), and multidrug-resistant (MDR) isolates revealed to be resistant against three or more tested antimicrobial classes (shaded in grey).

Pathogen	Number of Isolates	Category/Number of Antimicrobial Classes towards Isolates Were Not Susceptible							
		Pan-Susceptible	Not Susceptible						
					MDR				
		0	1	2	3	4	5	6	7
*P. multocida*	345 (100%)	52 (15.07%)	159 (46.09%)	86 (24.93%)	37 (10.72%)	6 (1.74%)	4 (1.16%)	1 (0.29%)	0 (0%)
*M. haemolytica*	273 (100%)	33 (12.09%)	176 (64.47%)	50 (18.32%)	11 (4.03%)	3 (1.10%)	0 (0%)	0 (0%)	0 (0%)

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
