# Peer review of "Antimicrobial Resistance in Isolates from Cattle with Bovine Respiratory Disease in Bavaria, Germany"

_antibiotics, 2021, doi:10.3390/antibiotics10121538_

Round 1
Reviewer 1 Report
The manuscript „ Antimicrobial Resistance in Isolates from Cattle with Bovine Respiratory Disease in Bavaria, Germany“ is a comprehensive an well written study. Study describes and discusses the current state on antimicrobial resistance of bacterial pathogens, common causative agents of bovine respiratory disease. Manuscript also reveals the impact of epidemiological parameters on antimicrobial resistance, such as farm type or number of animals. All the obtained results are clearly explained and summarized in provided Figures, Tables and Supplementary materials. There are few comments:
line 221_222_Figure 1 presents MDR rate of P. multocida over the five-year period. The Figure should contain a brief descriptive title. Note in the figure legend, what dashed lines and solid line present. It would be also helpful to add a p value in the Figure caption. P. multocida should be written in italic.
Line 226 Figure 2. Figure should contain a brief title.
Line 260, Figure 3 - Figure should contain a brief title. Label figure panels with letters a and b, as it is explained in the text and in the Figure caption. It would be helpful to add a p value in Figure caption
Reviewer 2 Report
This is a nice study reflecting the detection of major respiratory pathogens particularly bacteria such such as Pasteurella, Mannheimia, Tryperella etc from putative bovine respiratory disease syndrome along with their resistance profile trends over the period of 2015 to 2020 in Germany. In addition, some epidemiological factors wear also investigated. Some of the findings are very interesting and supported by methodology and results presented here. However, I have few comments that need to be address to make the findings more acceptable and understandable.
Its gentamicin not gentamycin (i not y), please update all…
No need to write the name of organism in FULL all the time, one its written first, them write it in brief as follows
Pasteurella multocida as P. multocida
Mannheimia haemolytica as M. haemolytica
Bibersteinia trehalosi as B trehalosi etc…..
What was the basis of selecting those antibiotics for the sensitivity test?
Define what is “pan-susceptible,”
Make a pie diagram to show the overall %/proportion of various pathogens detected among the total sample analyzed…
What is the difference between dairy farm, fattening farm and mixed farm management..define..??
Please make graph for other pathogen as done for Pasteurella in Figure 1.
I suggest to do a PCA analysis to make the findings more robust.
Reviewer 3 Report
- The genus name spelled as Tryperella needs to be corrected throughout the paper as Truperella.
- Line 32: Change incidences to incidence.
- Line 58: Change trimetoprim to trimethoprim.
- Line 65: Define what MDR means.
- Table 2: The authors list non-susceptible data to many antibiotics that do not have CLSI-endorsed clinical breakpoints, and therefore non-susceptible can not be defined. For example for P. multocida, the drugs Amoxicillin-clavulanate, cephalothin, trimethoprim-sulfamethoxazole, colistin, tiamulin, erythromycin, tilmicosin and gentamicin DO NOT have CLSI clinical breakpoints...so how can Susceptible, Intermediate and Resistance (Non-Susceptible) be determined? This is a very major error for the paper to list these values, and the authors need to remove this organism-drug combinations from the table. Additionally for M. haemolytica, the drugs Amox-Clav, cephalothin, TMP-SMX, colistin, tiamulin, erythromycin and gentamicin DO NOT have clinical breakpoints in order to define S,I,R and so the authors need to remove these organism-drug combinations from the table. Finally, there are absolutely no clinical breakpoints available for any of the drugs for T. pyogenes or B. trehalosi and so these organisms need to be removed from this Table.
- 6. Table 2: Change fluorochinolons to fluoroquinolones and tetracyclin to tetracycline.
- Table 3: There are no CLSI-endorsed clinical breakpoints for P. multocida with gentamycin and so S,I,R can not be determined and thus any statistical analysis can not be reported here...the authors need to remove this organism-drug combination from the table.
- Line 211: The authors may want to consider using the reference for definitions of MDR in veterinary medicine: Michael T Sweeney, Brian V Lubbers, Stefan Schwarz, Jeffrey L Watts, Applying definitions for multidrug resistance, extensive drug resistance and pandrug resistance to clinically significant livestock and companion animal bacterial pathogens, Journal of Antimicrobial Chemotherapy, Volume 73, Issue 6, June 2018, Pages 1460–1463, https://doi.org/10.1093/jac/dky043
- Figure 1: There are no clinical breakpoints available for any of the drugs for T. pyogenes or B. trehalosi and so these organisms need to be removed from this Figure.
- Line 542: Change microbouillon to microbroth.
Round 2
Reviewer 3 Report
Based on the author responses I am in agreement that the manuscript has been sufficiently improved to warrant publication in Antibiotics.
Thank you for the opportunity to review this important manuscript.